# Impact of Maximum Tongue Pressure in Patients with Jaw Deformities Who Underwent Orthognathic Surgery

**DOI:** 10.3390/diagnostics12020404

**Published:** 2022-02-04

**Authors:** Koichi Koizumi, Tomoaki Shintani, Yuki Yoshimi, Mirai Higaki, Ryo Kunimatsu, Yukio Yoshioka, Kazuhiro Tsuga, Kotaro Tanimoto, Hideki Shiba, Shigeaki Toratani

**Affiliations:** 1Department of Molecular Oral Medicine and Maxillofacial Surgery, Graduate School of Biomedical and Health Sciences, Hiroshima University, Hiroshima 734-8553, Japan; kkoizumi@hiroshima-u.ac.jp (K.K.); yyosioka@hiroshima-u.ac.jp (Y.Y.); tora@hiroshima-u.ac.jp (S.T.); 2Center of Oral Clinical Examination, Hiroshima University Hospital, Hiroshima 734-8551, Japan; bashihi@hiroshima-u.ac.jp; 3Department of Orthodontics and Craniofacial Developmental Biology, Graduate School of Biomedical and Health Sciences, Hiroshima University, Hiroshima 734-8553, Japan; yukimihsoy@hiroshima-u.ac.jp (Y.Y.); ryoukunimatu@hiroshima-u.ac.jp (R.K.); tkotaro@hiroshima-u.ac.jp (K.T.); 4Department of Oral and Maxillofacial Surgery, Hiroshima University Hospital, Hiroshima 734-8551, Japan; mirai-higaki@hiroshima-u.ac.jp; 5Department of Advanced Prosthodontics, Graduate School of Biomedical and Health Sciences, Hiroshima University, Hiroshima 734-8553, Japan; tsuga@hiroshima-u.ac.jp; 6Department of Biological Endodontics, Graduate School of Biomedical and Health Sciences, Hiroshima University, Hiroshima 734-8553, Japan

**Keywords:** maximum tongue pressure, orthognathic surgery, jaw deformity, malocclusion, sagittal split ramus osteotomy

## Abstract

Malocclusion and morphological abnormalities of the jawbone often affect the stomatognathic function and long-term postoperative stability in patients with jaw deformities. There are few reports on the effect of maximum tongue pressure (MTP) in these patients. We investigated the relationship between the MTP and jawbone morphology and the effect of the MTP on surgery in 42 patients with jaw deformity who underwent surgical orthodontic treatment at Hiroshima University Hospital. The MTP was measured using a tongue pressure measurement device; the average value was considered as the MTP. Based on the MTP measured before surgery, patients were classified into the high- or the low-MTP group. The clinical findings and results of the cephalometric analysis were compared. Posterior movement of the mandible in the high-MTP group was significantly lower than that in the low-MTP group. The ANB angle, overjet, and overbite in the high-MTP group were significantly smaller than those in the low-MTP group. On the other hand, there was no difference between the two groups in the measured values, indicating a labial inclination of the anterior teeth (U1 to SN, U1 to FH, IMPA, and FMIA). MTP has been suggested to affect mandibular prognathism in patients with jaw deformities.

## 1. Introduction

In addition to malocclusion and morphological abnormalities of the jawbone, patients with jaw deformities often show stomatognathic function [1,2,3]. In addition, in some cases, relapse is observed after surgical orthodontic treatment [4]. The causes of relapse include postoperative changes in the running of the masticatory muscles, the effects of soft tissues around the oral cavity, such as the lips and tongue, unstable occlusions, inadequate postoperative orthodontic treatment, and tongue protrusion habits [5]. Kobayashi et al. reported that in patients with skeletal mandibular prognathism, the extent of the relapse tends to increase as the amount of posterior movement of the mandible increases [6]. Therefore, when performing the surgical orthodontic treatment, it is important to observe the condition of the maxillofacial muscles and soft tissues around the oral cavity. To ensure long-term postoperative stability, it is necessary to examine the stomatognathic functions of patients undergoing orthognathic surgery. Previous studies have reported that surgical orthodontic treatment improves masticatory ability, occlusal force, mandibular condyle and lip movement, and muscle activity [7,8,9]. In particular, not only tongue size or tongue habit but also tongue adaptation to changes in the volume of the oral cavity after the orthodontic surgery have been reported to have a great influence on treatment success [10].

Measurements of tongue pressure include resting pressure, chewing pressure, swallowing pressure, pronunciation pressure, and maximum tongue pressure (MTP). Video recording of tongue movement or tongue pressure measurement was performed to evaluate tongue function [11,12]. Previous studies reported the use of a small pressure sensor such as a flush diaphragm pressure transducer, which is placed in a palatal appliance or a replica denture [13]. This sensor is expensive and cannot be sterilized, so it cannot be used in many patients or postoperative patients. Utanohara et al. developed a measurement device for measuring MTP using a balloon-type disposable oral probe designed for clinical use [14]. Using this device, those authors also established standard MTP values based on subject gender and age, who had no history of dysphagia and maintained occlusal contact [14]. It has been reported that tongue pressure is significantly lower in patients with dysphagia as compared with controls [15]. A recent report showed that the MTP in 10 patients after orthognathic surgery was lower than that of patients with normal occlusion [16]. Other reports showed that MTP tends to increase up to 1 year after surgery [17]. No reports are investigating the effect of MTP on the jawbone growth or labial tilt of the anterior teeth in patients with jaw deformity. In addition, there are few reports on changes in MTP before and after orthognathic surgery [18].

This is the first report to investigate whether MTP affects the overgrowth of the mandible or the labial tilt of the anterior teeth in patients with jaw deformities who underwent orthognathic surgery to reposition the mandible. Furthermore, changes in MTP due to surgery were monitored for up to a year after the surgery.

## 2. Materials and Methods

### 2.1. Study Population

We enrolled in this study 42 consecutive patients with jaw deformities diagnosed as skeletal class II or III at the Department of Orthodontics, Hiroshima University Hospital, with or without maxillary deformity, who underwent orthognathic surgery between January 2014 and December 2015 at the Department of Oral and Maxillofacial Surgery, Hiroshima University Hospital. Baseline clinical characteristics, including age, sex, body mass index, surgical methods, overbite (at the first visit), overjet (at the first visit), movement of the mandible at surgery, preoperative MTP, and postoperative (up to 12 months) MTP were recorded. We analyzed conventional lateral cephalometric measurements preoperatively, based on a previous report. This study was approved by the research ethics board of Hiroshima University (approval no. epidemiology—2487). Written informed consent was obtained from all patients. All procedures were performed in accordance with the ethical standards of the institutional and/or national research committee and with the Helsinki Declaration and its later amendments or comparable ethical standards.

### 2.2. MTP Measurements

To test the MTP, we used the JMS tongue pressure measurement instrument (TPM-01, JMS Co. Ltd., Hiroshima, Japan). MTP was measured using a disposable probe and a simple pressure-recording manometer. By pushing the pressurization button in the body of TPM-01, the probe was inflated with air at an initial pressure of 19.6 kPa. This pressure was taken as zero for calibration. Pressures were recorded with participants seated comfortably in an upright position during all measurements. Participants were asked to place a plastic pipe lightly between the upper and lower incisors to stabilize the balloon parts in the oral cavity. During all measurements, the participants held the cylinder with their incisors such that the balloon could be placed between the tongue and the anterior section of the palate. Using the probe, we measured the pressure of the front part of the tongue, especially the tongue tip, exerted by the extrinsic and intrinsic muscles of the tongue against the hard palate. Participants were also asked to close their lips. Then, the examiner asked the participants to compress the balloon onto the palate for 7 s with maximum voluntary effort. Consequently, we measured the tongue pressure with the participant’s mouth slightly open, which may have affected the accuracy of the measurement. Measurements were performed three times for each participant. The participants were given a rest period of about 30 s between measurements, during which they rinsed their mouths to ensure their oral conditions were uniform. The median value of the three measurements was defined as the representative MTP.

MTP was measured (1) preoperation, (2) 1–3 months after the operation, (3) 4–6 months after the operation, (4) 7–9 months after the operation, and (5) 10–12 months after the operation.

### 2.3. Statistical Analysis

Preoperative and postoperative MTP were compared using a paired t-test, and other comparisons were performed using the Mann–Whitney *U*-test. The significance of differences in categorical variables (gender) was calculated using the chi-squared test. Spearman’s rank correlation coefficient was calculated to explore the relation between MTP and overbite, overjet, and mandible movement at the surgery. Statistical analysis was performed using JMP 14 statistical software (SAS Institute Inc., Cary, NC, USA). The differences were considered significant at *p* < 0.05.

## 3. Results

A total of 42 patients (12 males, 30 females; 14–44 years) with jaw deformities who underwent orthognathic surgery were registered in this study. Table 1 shows the baseline characteristics of all patients. All participants underwent sagittal split ramus osteotomy (SSRO; according to Dal Pont–Obwegeser method) for setback or advancement of the mandible (including 17 patients who underwent Le Fort I osteotomy as well as SSRO) [19,20]. Two patients underwent tongue reduction. For these patients, the median overbite, overjet, and movement of the mandible at SSRO (interquartile range (IQR)) were 0 (−2.0, 1.9) mm, 0.3 (−2.2, 1.9) mm, and −7.0 (−8.9, −4.9) mm, respectively. The median preoperative MTP (IQR) was 32.3 (28.5, 48.0) kPa.

We divided the subjects into low preoperative MTP (male: <35 kPa; female: <30 kPa) and high preoperative MTP (male: ≥35 kPa, female: ≥30 kPa) groups. Table 1 shows the univariate analysis of baseline characteristics for comparing the clinical parameters between the two groups. Median values of overjet (high vs. low; −0.65 mm vs. 1.8 mm) and overbite (−0.35 mm vs. 1.1 mm) were significantly less in the high-MTP group than in the low-MTP group (*p* < 0.05). The movement of the mandible (−8.5 mm vs. −2.75 mm) was significantly greater in the high-MTP group than in the low-MTP group (*p* < 0.05).

We next investigated cephalometric changes in the two groups. As a result, the SNA (high vs. low in median value; 79.9° vs. 81.8°) and ANB angles (−0.8° vs. 1.15°) of the high-MTP group were significantly smaller than those of the low-MTP group (*p* < 0.05).

Results of the linear regression analysis revealed weak negative correlations between preoperative MTP and overbite (Figure 1A; *r* = −0.27), overjet (Figure 1B; *r* = −0.27), and mandibular movement (Figure 1C, *r* = −0.28).

Finally, we compared preoperative and postoperative MTP. The group of patients with high preoperative MTP also had high postoperative MTP (Table 1). The MTP at 1–3 months postoperation was significantly lower than the MTP during other periods (*p* < 0.05; Figure 2, Table 2).

## 4. Discussion

Although there are reports of decreased tongue pressure associated with oral hypofunction in older adults, there are few reports on tongue pressure in young people, especially those with jaw deformities who have undergone surgical orthodontic treatment [21,22]. This study investigated the relationship between MTP and jawbone morphology and the effect of MTP on surgery. MTP exhibits a weak positive correlation with BMI (*r* = 0.25) and is considered to contribute to the error in measured values. The MTP was slightly lower in the subjects of this study as compared with that reported by Utanohara et al. [14]. We speculate that the occlusion was unstable immediately after preoperative orthodontic treatment. Utanohara et al. also reported that the MTP of males was significantly higher than that of females in young people (i.e., the 20 s, 30 s, and 40 s); thus, we defined high MTP as an MTP of ≥35 kPa for men and ≥30 kPa for women and low MTP as <35 kPa for men and <30 kPa for women [14]. We demonstrated that overjet and overbite were significantly less in the high-MTP group than in the low-MTP group. The amount of movement of the mandible was significantly greater in the high-MTP group than in the low-MTP group. In addition, there was a weak correlation between MTP and these clinical parameters. Based on these results, we considered that the mandible and/or anterior teeth were pushed forward by high MTP. The cephalometric analysis revealed that the SNA and ANB angles in the high-MTP group were significantly larger than those in the low-MTP group. The ANB angle in the high-MTP group was significantly smaller than that in the low-MTP group, indicating that the mandible in the high-MTP group was located more anteriorly to the maxilla than that in the low-MTP group. We also found that the SNA angle in the high-MTP group was significantly smaller than that in the low-MTP group, but the reason for this is not clear. On the other hand, we found no difference between the two groups in the measured values, indicating a labial inclination of the anterior teeth (U1 to SN, U1 to FH, IMPA, and FMIA). A prior study reported no difference in MTP among the Angle Classes or between males and females [23]. Another report showed that the pressures generated on the lingual surface against the lower teeth were about twice as heavy as those on the labial surface [24]. We believe our results are important because MTP affects the degree of overgrowth of the mandible in patients with jaw deformities. In particular, the risk of mandibular anterior collision can be further increased if patients with high MTP have a tongue protrusion habit [25].

In terms of changes in MTP over time, MTP at only 1–3 months after surgery was significantly lower than MTP in other periods. When measuring MTP, the probe is placed between the tongue and the anterior teeth of the plate. After surgery, the entire tongue moves posteriorly (or anteriorly), and the distance between the tongue and the anterior teeth of the plate changes, giving an impression that MTP decreases temporarily. Previous studies on oral function in patients with jaw deformities have reported that the MTP changes within a few months after surgery to adapt to the new postoperative environment [16,26,27]. Another report indicated that MTP after SSRO tends to increase over 1 year after surgery. It is believed that this is due to the difference in the tongue pressure–measuring device used and the number of cases [17].

Because no patient experienced relapse at up to 1 year, one limitation of this study is that the relationship between MTP and relapse needs to be investigated by long-term follow-up rather than merely a year after surgery. By including nonsurgical cases, the effect of MTP could be better clarified. Furthermore, the symptoms of the temporomandibular joint may appear because the mandibular condyle is displaced anteriorly due to high MTP [28]. It is also debatable whether high tongue pressure is involved in the removal of composite resin, one of the dental restorations [29].

## 5. Conclusions

In this study, we investigated the impact of the MTP in patients with jaw deformities who underwent orthognathic surgery. Our findings revealed that MTP affects the anterior protrusion of the mandible rather than the labial inclination of the anterior teeth (Figure 3). Postoperative MTP decreased for up to three months, after which no significant difference was observed compared to preoperative MTP. Therefore, it can be inferred that MTP influences the prognosis of orthodontic treatment, and measuring the MTP of patients undergoing orthodontic treatment is deemed relevant.

## Figures and Tables

**Figure 1 diagnostics-12-00404-f001:**
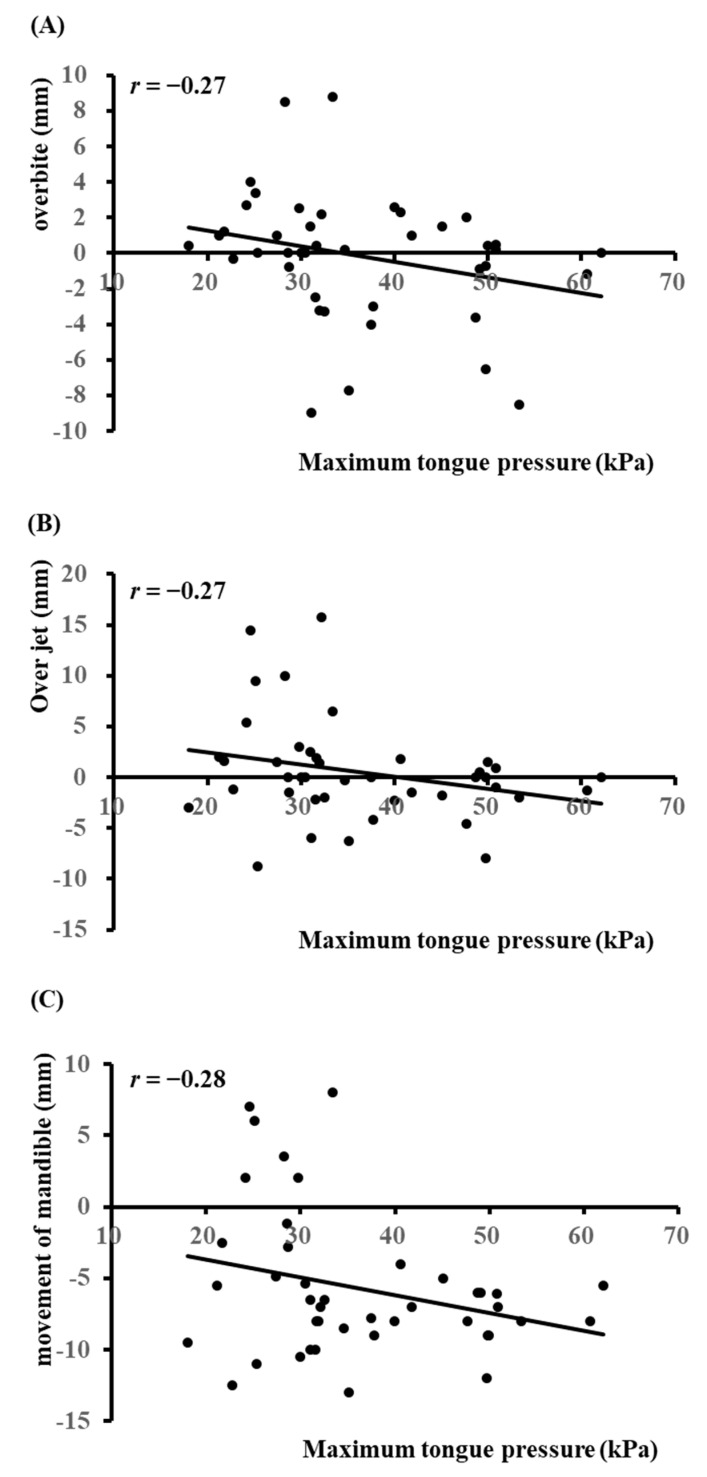
Linear regression analysis between maximum tongue pressure and overbite (**A**), overjet (**B**), and movement of the mandible (**C**).

**Figure 2 diagnostics-12-00404-f002:**
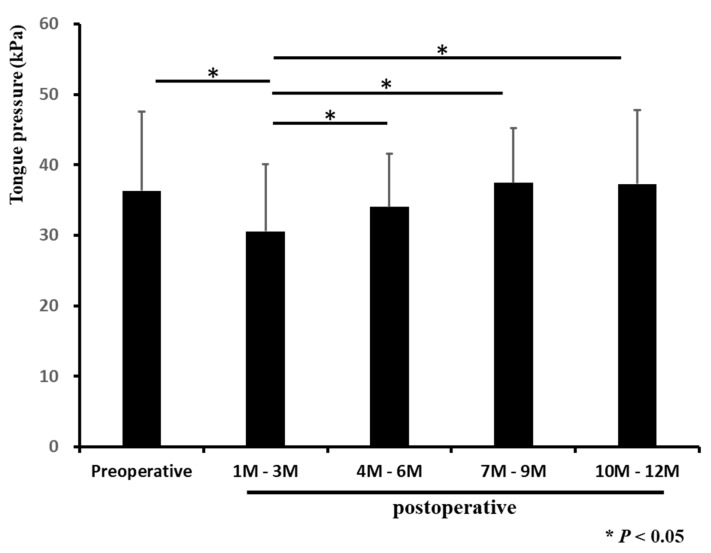
Changes in maximum tongue pressure before and after surgery. MTP was measured before and one year after surgery. The MTP at three months after the operation was significantly lower than that in other periods. Each bar represents the mean ± SD. * *p* < 0.05.

**Figure 3 diagnostics-12-00404-f003:**
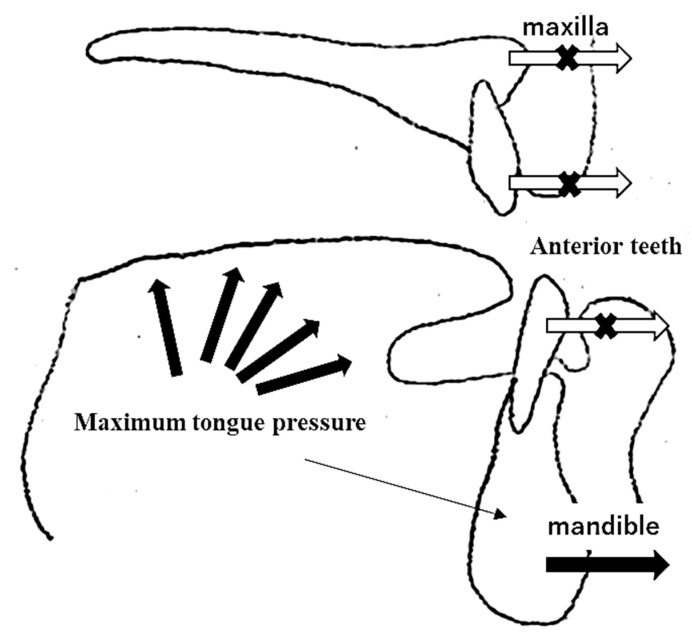
The scheme of this study. The position of the mandible was affected by high MTP.

**Table 1 diagnostics-12-00404-t001:** Univariate analysis of baseline characteristics to investigate the impact of maximum tongue pressure.

	Total (*n* = 42)	Low Tongue Pressure ^b^ (*n* = 15)	High Tongue Pressure ^c^ (*n* = 27)	*p* Value (Low vs. High)
Age (years), median (QRT)	20 (18, 28.3)	25 (17.5, 33.5)	20 (18, 21.5)	0.1
Gender ^a^ (Male/Female)	12/30	5/12	7/18	0.92
Body mass index (kg/mm^2^), median (QRT)	20.6 (19.3, 22.1)	20.7 (19.4, 21.8)	20.6 (18.7, 22.7)	0.94
Over jet (mm), median (QRT)	0.3 (−2.2, 1.9)	1.8 (−1.43, 6.23)	−0.65 (−2.15, 0.8)	0.04 *
Overbite (mm), median (QRT)	0 (−2.0, 1.9)	1.1 (0, 3.23)	−0.35 (−3.28, 0.88)	0.01 *
Amount of movement ^d^ (mm), median (QRT)	−7 (−8.9, −4.9)	−2.75 (−9.5, 2)	−8.5 (−8, −6.1)	0.03 *
Surgical method (cases)				
SSRO	42			
LeFort I	17			
tongue reduction	2			
Preoperative tongue pressure (kPa), median (QRT)	32.3 (28.5, 48.0)	27.2 (23.1, 30.9)	42.1 (34.3, 50)	<0.001
Postoperative tongue pressure (kPa), median (QRT)				
1M–3M (*n* = 35)	29.6 (23.2, 37.4)	23 (17.3, 32.2)	32.9 (28.3, 40.6)	0.008 *
4M–6M (*n* = 21)	33.4 (30.8, 37.7)	33.1 (30.4, 36.8)	37.4 (32.3, 42.2)	0.07
7M–9M (*n* = 17)	40.1 (32.4, 44.1)	34.7 (23.3, 41.9)	41.4 (34.5, 44.1)	0.15
10M–12M (*n* = 22)	36.8 (31.5, 44.0)	31.9 (24.6, 36.2)	42 (35.7, 49.2)	0.009 *
Cephalometric analysis				
gonial angle (degree), median (QRT)	127 (122, 134)	126 (117, 135)	127 (123, 133)	0.52
SNA (degree), median (QRT)	81 (79, 83)	81.8 (80, 84.5)	79.9 (78.3, 82.2)	0.04 *
SNB (degree), median (QRT)	82 (76, 84)	80.6 (74.6, 83)	81.7 (77.2, 85.1)	0.29
ANB (degree), median (QRT)	−0.2 (−2.8, 2.2)	1.15 (−2.35, 8.08)	−0.8 (−2.75, 0.7)	0.03 *
Interincisal angle (degree), median (QRT)	128 (117, 136)	120 (109, 138)	132 (118, 136)	0.37
U1 to SN (degree), median (QRT)	105(99, 111)	105 (97, 110)	105 (101, 112)	0.64
U1 to FH (degree), median (QRT)	115 (111, 121)	115 (107, 125)	115 (112, 119)	0.8
FMA (degree), median (QRT)	31 (27, 35)	31.3 (25.8, 37)	30.5 (26.8, 33.8)	0.71
IMPA (degree), median (QRT)	84 (79, 90)	85 (75.5, 92.5)	83.3 (79.8, 89.6)	1
FMIA (degree), median (QRT)	63 (58, 70)	63.2 (49.2, 66.1)	64.3 (59, 74.9)	0.24

Test used for analysis: Wilcoxon rank-sum test. ^a^ Test used for analysis: *χ*^2^ test. ^b^ Low tongue pressure: male <35 kPa, female <30 kPa. ^c^ High tongue pressure: male ≥35 kPa, female ≥30 kPa. ^d^ Minus means moving backward. QRT, quartile; SSRO, sagittal split ramus osteotomy; LeFort I, LeFort I maxillary osteotomy. * *p* < 0.05.

**Table 2 diagnostics-12-00404-t002:** *p*-value by comparison of the MTP in each period.

	Preoperative	1–3M	4–6M	7–9M	10–12M
Preoperative		<0.01	0.65	0.78	0.12
1–3M			<0.01	<0.01	<0.01
4–6M				0.97	0.08
7–9M					0.38
10–12M					

Test used for analysis: paired *t*-test.

## Data Availability

Data used in this study are openly available in a public repository that issues data sets with DOIs.

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
