# Peer review of "Impact of Maximum Tongue Pressure in Patients with Jaw Deformities Who Underwent Orthognathic Surgery"

_diagnostics, 2022, doi:10.3390/diagnostics12020404_

Round 1

Reviewer 1 Report

There are some weaknesses through the manuscript which need improvement. Therefore, the submitted manuscript cannot be accepted for publication in this form, but it has a chance of acceptance after a minor revision. My comments and suggestions are as follows:

1- Abstract gives information on the main feature of the performed study, but a couple of sentence about background of the study must be added.

2- Authors must clarify necessity of the performed research. Objectives of the study must be clearly mentioned in introduction.

3- The literature study must be enriched. In this respect, authors must read and refer to the following papers: (a) https://doi.org/10.1016/j.adaj.2019.07.034 (b) https://doi.org/10.1016/j.jmbbm.2019.02.009 and other related papers.

4- It would be nice, if authors could add some figures (real or schematic) to show concept and some conditions.

5- Reason on the deviations (Fig. 2) must be discussed. In addition, error in measurement must be considered and discussed.

6- In its language layer, the manuscript should be considered for English language editing. There are sentences which have to be rewritten.

7- The conclusion must be more than just a summary of the manuscript. List of references must be updated based on the proposed papers. Please provide all changes by red color in the revised version.

Reviewer 2 Report

No further suggestions for authors.

Author Response

Thank you very much for the review. Our manuscript was edited by the language service.

Reviewer 3 Report

I would like to congratulate authors for addressing this interesting matter and for managing to elaborate a comprehensive analysis. I especially like to congratulate the authors for managing to obtain results that can be clinically significant and could lead to change in treatment modalities. 

Here are a few suggestions on my behalf:  I consider important for the authors to offer more details regarding the orthognatic treatment the patients underwent and if the subpopulation undergoing surgery is homogeneous.    Moreover, I consider opportune to elaborate a bit on the possible clinical implications of the present article's findings. The conclusion that ' postoperative follow-up should be performed more carefully in patients with high-MTP' is a bit scarce.  
